# SEQUENTIAL TEST-TIME ADAPTATION VIA MARTINGALE-DRIVEN FISHER PROMPTING

## ABSTRACT

We present a theoretical framework for M-FISHER, a method for sequential distribution shift detection and stable adaptation in streaming data. For detection, we construct an exponential martingale from non-conformity scores and apply Ville's inequality to obtain time-uniform guarantees on false alarm control, ensuring statistical validity at any stopping time. Under sustained shifts, we further bound the expected detection delay as $\mathcal{O}(\log(1/\delta)/\Gamma)$, where $\Gamma$ reflects the post-shift information gain, thereby linking detection efficiency to distributional divergence. For adaptation, we show that Fisher-preconditioned updates of prompt parameters implement natural gradient descent on the distributional manifold, yielding locally optimal updates that minimize KL divergence while preserving stability and parameterization invariance. Together, these results establish M-FISHER as a principled approach for robust, anytime-valid detection and geometrically stable adaptation in sequential decision-making under covariate shift.

## 1 INTRODUCTION

Recent advances in vision-language models (VLMs) such as CLIP (Radford et al., 2021) have transformed few-shot image recognition by leveraging large-scale pre-training on aligned image-text pairs. Despite their strong zero-shot performance, these models remain vulnerable when deployed in real-world environments, where test data often exhibit distribution shifts from the training distribution. Such shifts can lead to degraded accuracy due to covariate drift (Sugiyama et al., 2012) and exacerbate overconfidence in predictions (Guo et al., 2017), limiting their reliability in safety-critical applications.

A common strategy to improve robustness is *prompt tuning*, where learnable text embeddings are optimized to align image and text features more effectively. Methods such as CoOp (Zhou et al., 2022) and ProDA (Lu et al., 2022) have demonstrated strong gains under in-distribution conditions. However, these approaches are typically trained offline and produce static prompts, leaving them unable to adapt to evolving test-time shifts. Another line of work focuses on *test-time adaptation* (TTA) (?Liang et al., 2020), which dynamically updates model parameters during deployment. While effective for convolutional networks, these methods often lack theoretical grounding when applied to VLMs, and can overfit when test batches are small.

Beyond prompt tuning and TTA, several works aim to correct covariate shift using importance weighting or calibration-aware objectives (Shi et al., 2021; Khan et al., 2025). For instance, prior work by Khan et al. (2024) introduced Fisher information-based penalties to improve calibration under distribution shift. However, these methods do not account for *sequential* shift patterns that emerge in streaming data, and thus cannot determine *when* adaptation should occur, often leading to either delayed response or excessive, noisy updates.

In this work, we propose M-FISHER, a novel TTA framework that unifies sequential shift detection with theoretically principled prompt adaptation. Our key insight is to leverage martingale-based change detection (Balsubramani & Ramdas, 2015) to provide rigorous, data-dependent triggers for adaptation, ensuring updates occur only when statistically significant distribution shifts are detected. Upon detection, we adapt prompts using natural gradient updates preconditioned by the Fisher Information Matrix (Khan et al., 2024), which scales updates in parameter space to mitigate instability and overfitting. This combination yields a framework with both strong empirical adaptability and theoretical guarantees for false alarm control and adaptation stability.

**Contributions.** The main contributions of this work are:

- We propose M-FISHER, a novel framework for *sequential* test-time adaptation of vision-language models (VLMs) that unifies distribution shift detection and prompt adaptation.

- We introduce a *martingale-based change detector* with time-uniform false alarm guarantees, enabling adaptation to be triggered only when statistically significant shifts occur.

- We develop *Fisher-preconditioned prompt updates* that scale adaptation steps according to the local information geometry of the prompt parameter space, improving stability and preventing overfitting to transient noise.

- We provide theoretical analysis showing: (i) bounded false alarm probability and detection delay under sustained shifts, and (ii) local KL-optimality of Fisher-preconditioned updates.

- We demonstrate state-of-the-art sequential adaptation performance on multiple domain shift benchmarks, with robustness to mixed and evolving test streams.

## 2 RELATED WORK

**Prompt Tuning for Vision-Language Models.** Prompt tuning methods adapt the textual input of vision-language models (VLMs) to better align with the visual encoder's representation space. CoOp (Zhou et al., 2022) learns continuous prompt embeddings for each class, while ProDA (Lu et al., 2022) regularizes prompts through diversity and uncertainty modeling. Although these approaches achieve strong gains on few-shot and domain-generalization benchmarks, they produce static prompts learned offline. Consequently, they cannot respond to evolving distribution shifts during deployment.

**Test-Time Adaptation.** Test-time adaptation (TTA) updates model parameters on-the-fly to reduce performance degradation under distribution shift. Tent (Wang et al., 2020) minimizes entropy on test samples, while SHOT (Liang et al., 2020) leverages self-training with pseudo-labels. Most existing methods are designed for convolutional networks and rely on continuous updates without explicit shift detection, which can lead to overfitting, especially when test batches are small or contain mixed distributions. Recent works on VLM adaptation have primarily focused on static fine-tuning rather than sequential adaptation.

**Distribution Shift Detection.** Detecting when a model's input distribution changes is a long-studied problem in statistics and sequential analysis. Martingale-based methods (Balsubramani & Ramdas, 2015) offer rigorous guarantees on false alarm probability and have been used in domains such as finance and online monitoring. In the context of machine learning, shift detection has been explored through conformal prediction (Filos et al., 2020), feature space divergence (Rabanser et al., 2019), and test statistics (Gretton et al., 2009). However, these methods are typically decoupled from the adaptation process, requiring manual intervention once a shift is detected.

**Fisher Information in Model Adaptation.** The Fisher Information Matrix (FIM) captures local curvature of the loss landscape and has been used for parameter regularization in continual learning (Kirkpatrick et al., 2017) and for natural gradient optimization (Amari, 1998). In prompt tuning, Fisher-based updates have been shown to stabilize adaptation (Khan et al., 2024). Yet, prior approaches have applied Fisher penalties in an offline or batch adaptation setting, without integrating them into a sequential shift detection framework.

## 3 METHOD

We introduce **M-FISHER**, a sequential test-time adaptation framework that couples divergence-aware martingale shift detection with Fisher information-guided prompt updates. This approach is designed for vision-language models (VLMs) such as CLIP, and addresses covariate shift in streaming test data.

## 3.1 PRELIMINARIES

Let $f_\theta$ denote a VLM with image encoder $f_v$, text encoder $f_t$, and learnable prompt embeddings $P = [p_1, \ldots, p_k]$. At deployment, the model processes a stream $\{x_t\}_{t=1}^T$ where the input distribution may drift over time.

We define two quantities central to our framework:

- **Non-conformity score**:

$$S_t = \mathrm{KL}(p_\theta(y|x_t) \,\|\, \mathrm{Uniform}) + \alpha \, \|f_v(x_t) - \mu_{\mathrm{train}}\|_{\Sigma^{-1}}^2, \tag{1}$$

  where the first term measures confidence deviation from uniform, the second is the Mahalanobis distance in image feature space, and $\alpha$ is a scaling hyperparameter.

- **Prompt-level Fisher Information Matrix (FIM)**:

$$F_P = \mathbb{E}_{x \sim \mathcal{D}_{\mathrm{train}}} \big[ \nabla_P \log p_\theta(y|x) \, \nabla_P \log p_\theta(y|x)^\top \big], \tag{2}$$

  where gradients are taken only with respect to $P$, with the backbone frozen.

## 3.2 MARTINGALE SHIFT DETECTION

We adopt an exponential supermartingale $\{M_t\}$ to detect distribution shifts in a statistically controlled manner. Let $\hat{\mu} = \frac{1}{n} \sum_{j=1}^n S_j^{\mathrm{cal}}$ be the mean score computed on a held-out calibration set $\mathcal{C} = \{S_j^{\mathrm{cal}}\}_{j=1}^n$. The martingale is updated as:

$$M_t = M_{t-1} \cdot \exp\big(\lambda(S_t - \hat{\mu}) - \psi(\lambda)\big), \quad M_0 = 1, \tag{3}$$

where $\psi(\lambda) = \log \mathbb{E}_{\mathrm{cal}}[\exp(\lambda(S_t - \hat{\mu}))]$ ensures $\mathbb{E}[M_t] \leq 1$ under the no-shift hypothesis.

**Finite-sample correction.** With finite calibration data, plug-in estimates of $\psi(\lambda)$ may underestimate the true moment generating function, breaking the supermartingale property. To restore validity we compute a *high-confidence upper bound* $\bar{\psi}(\lambda)$ using a nonparametric bootstrap. Specifically, for each $\lambda$ we compute $\widehat{m}(\lambda) = \frac{1}{n} \sum_{j=1}^n e^{\lambda(S_j^{\mathrm{cal}} - \hat{\mu})}$. From $B$ bootstrap resamples $\mathcal{C}^{(b)}$, we obtain $\widehat{m}^{(b)}(\lambda)$ and take the empirical $(1 - \alpha)$-quantile $m_{(1-\alpha)}(\lambda)$. We then set

$$\bar{\psi}(\lambda) = \log m_{(1-\alpha)}(\lambda).$$

The corrected process is

$$M_t = \prod_{i=1}^t \exp\big(\lambda(S_i - \hat{\mu}) - \bar{\psi}(\lambda)\big). \tag{4}$$

Conditional on the calibration event $\{\psi(\lambda) \leq \bar{\psi}(\lambda)\}$, $M_t$ is a supermartingale and Ville's inequality ensures

$$\mathbb{P}\left(\sup_{t \geq 1} M_t \geq \tau \,\Big|\, \mathcal{C}\right) \leq \tfrac{1}{\tau}. \tag{5}$$

Unconditionally, the false alarm probability is bounded by $\alpha + 1/\tau$. When $M_t \geq \tau$, we declare a shift and trigger prompt adaptation.

## 3.3 FISHER PROMPT ADAPTATION

Upon detection, we update prompts via a natural gradient step:

$$\nabla_P^{\mathrm{Fisher}} = F_P^{-1} \nabla_P\big(S_t + \mathcal{L}_{\mathrm{CMP}}\big), \tag{6}$$

$$P \leftarrow P - \eta \nabla_P^{\mathrm{Fisher}}, \tag{7}$$

where $F_P^{-1}$ is approximated using a damped diagonal estimator (Kunstner et al., 2019) for computational efficiency.

### 3.4 CONFIDENCE RECALIBRATION

To improve post-adaptation reliability, we incorporate a *confidence misalignment penalty*:

$$\mathcal{L}_{\mathrm{CMP}} = \widetilde{\mathrm{ECE}}\big(p_\theta, \mathcal{D}_{\mathrm{cal}}\big), \tag{8}$$

where $\widetilde{\mathrm{ECE}}$ is a differentiable approximation to the Expected Calibration Error, computed over a calibration buffer. This term regularizes the natural gradient updates and mitigates overconfident predictions.

**M-FISHER summary.** M-FISHER integrates (i) sequential shift detection via e-process martingales, and (ii) prompt adaptation via Fisher-preconditioned updates. At each time $t$, we compute a non-conformity score $S_t$ (Eq. 1) and update the e-process $M_t$ (Eq. 4). A shift is declared when $M_t \geq \tau$, where $\tau = 1/\delta$ ensures false alarm probability at most $\alpha + \delta$. Upon detection, prompt parameters $P$ are updated using Fisher-preconditioned natural gradients (Eq. 7) with loss

$$\mathcal{L}_t = S_t + \mathcal{L}_{\mathrm{CMP}}.$$

The overall mechanism of our method is illustrated in Figure 3.4.

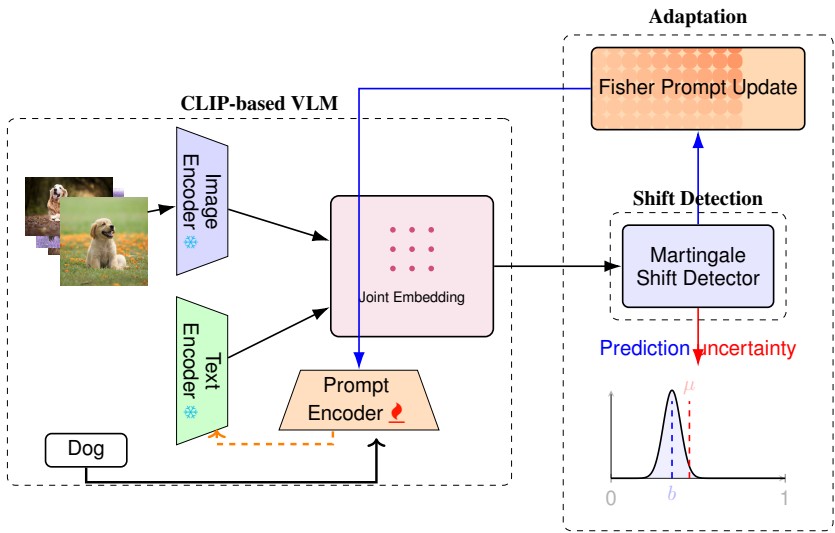

Figure 1: Sequential shift detection and adaptation in M-FISHER. The prediction distribution (blue) is monitored via non-conformity scores. When the martingale statistic exceeds threshold $\tau$, Fisher-natural gradient updates stabilize prompt adaptation while minimizing KL divergence.

**Algorithm.** Algorithm 1 summarizes the procedure. Importantly, adaptation is triggered only after statistically valid detection, avoiding unnecessary overfitting to transient fluctuations.

---

**Algorithm 1** M-FISHER: Martingale-Driven Fisher Prompt Adaptation

---

**Require:** Test stream $\{x_t\}$, initial prompts $P$, threshold $\tau$, scaling $\alpha$, learning rate $\eta$
1: Compute $\mu_S$ and $\psi(\lambda)$ on calibration data
2: Initialize $M_0 = 1$
3: **for** $t = 1$ to $T$ **do**
4:     Compute $S_t$ via Eq. 1
5:     Update $M_t$ via Eq. 3
6:     **if** $M_t > \tau$ **then**
7:         Compute $\nabla_P^{\mathrm{Fisher}}$ via Eq. 6
8:         Update $P$ via Eq. 7
9:     **end if**
10: **end for**
11: **return** Adapted prompts $P$

---

## 4 THEORETICAL ANALYSIS

We provide formal justification for the two key components of **M-FISHER**: (1) sequential shift detection via martingale (e-process) statistics, and (2) stability of adaptation via Fisher-preconditioned updates.

- **Detection**: Eq. 5 ensures a $1/\tau$ bound on the false alarm probability, and Lorden's bound (Lorden, 1971) implies bounded detection delay under sustained shifts.
- **Adaptation**: By Amari's natural gradient theory (Amari, 1998), the update in Eq. 7 locally minimizes $\mathrm{KL}(p_{\theta,P_{\mathrm{train}}} \,\|\, p_{\theta,P_{\mathrm{test}}})$ in the prompt parameter space.

### 4.1 FALSE ALARM CONTROL VIA VILLE'S INEQUALITY

Let $\{S_t\}_{t=1}^{\infty}$ denote the non-conformity scores defined in Eq. 1, and let $\mu_S$ be their expectation under the null hypothesis $\mathcal{H}_0$ (no shift). Define the exponential process

$$M_t \;=\; \prod_{i=1}^{t} \exp\big(\lambda(S_i - \mu_S) - \psi(\lambda)\big), \qquad M_0 = 1, \tag{9}$$

where $\psi(\lambda) = \log \mathbb{E}_{\mathcal{H}_0}[\exp(\lambda(S_i - \mu_S))]$ is the log-moment generating function under $\mathcal{H}_0$. By construction, $\{M_t\}$ is a nonnegative supermartingale with $\mathbb{E}[M_t] \leq 1$ (**?**). Ville's inequality (**?**) then yields the time-uniform false alarm control

$$\mathbb{P}_{\mathcal{H}_0}\left(\sup_{t \geq 1} M_t \geq \tau\right) \leq \frac{1}{\tau}, \qquad \tau > 0, \tag{10}$$

which is valid at arbitrary stopping times and thus suitable for streaming deployment (Howard et al., 2021)see detail A.1.

### 4.2 DETECTION DELAY UNDER SUSTAINED SHIFT

Suppose a shift occurs at $t^\star$ so that the score distribution changes from $P_0$ to $P_1$. Define the stopping time

$$T(\tau) \;=\; \inf\{t \geq 1 : M_t \geq \tau\}.$$

For likelihood-ratio (LR) e-processes (i.e., $M_t$ equal to or lower-bounded by the cumulative LR), classical results (Lorden, 1971) give

$$\mathbb{E}_{P_1}\big[(T(\tau) - t^\star)^+\big] \;\lesssim\; \frac{\log \tau}{\mathrm{KL}(P_1\|P_0)}, \tag{11}$$

up to universal constants. In the more general case of an exponential e-process as in Eq. (1), the same scaling holds with $\mathrm{KL}(P_1\|P_0)$ replaced by the post-shift exponential growth rate.

$$\Gamma \;=\; \sup_{\lambda > 0}\Big\{\lambda\big(\mathbb{E}_{P_1}[S] - \mu_S\big) - \psi(\lambda)\Big\}, \tag{12}$$

so that $\mathbb{E}_{P_1}[(T(\tau) - t^\star)^+] \lesssim \log(\tau)/\Gamma$. Intuitively, larger distributional divergence (or growth rate) yields faster detection. This refines mean-difference heuristics and makes explicit the dependence on information divergence.

### 4.3 FISHER-PRECONDITIONED PROMPT UPDATES

Let $\mathcal{P}$ denote the prompt parameter subspace with coordinates $P \in \mathbb{R}^k$, and let $F_P$ be the (empirical) Fisher Information Matrix from Eq. 2. A Euclidean gradient step

$$P \leftarrow P - \eta \, \nabla_P \mathcal{L}$$

can be unstable under ill-conditioning. The natural gradient step

$$P \leftarrow P - \eta \, (F_P + \gamma I)^{-1} \nabla_P \mathcal{L}$$

(with Tikhonov damping $\gamma > 0$) instead performs steepest descent in the Fisher–Rao geometry (Amari, 1998; Kunstner et al., 2019). Equivalently, for small steps it minimizes the first-order approximation of

$$\mathrm{KL}\big(p_{\theta,P}(\cdot \mid x) \,\big\|\, p_{\theta,P+\Delta P}(\cdot \mid x)\big) \tag{13}$$

among perturbations $\Delta P$ achieving the same first-order decrease in $\mathcal{L}$. In **M-FISHER**, taking $\mathcal{L} = S_t + \mathcal{L}_{\mathrm{CMP}}$ yields updates that (i) are invariant to reparameterization, (ii) respect the local information geometry of prompts, and (iii) reduce overfitting to transient noise after a detected shift.

## 5 EXPERIMENTS

### 5.1 EXPERIMENTAL SETUP

We evaluate **M-FISHER** on benchmark datasets designed to test robustness under distribution shift, following the sequential test-time adaptation setting introduced in Wang et al. (2020).

**Datasets.** We evaluate M-FISHER on benchmarking datasets including ImageNet-C (**?**) with 15 corruption types at 5 severity levels applied to ImageNet validation images, ImageNet-R (Hendrycks et al., 2021) images from 30 classes rendered in diverse artistic styles (sketches, paintings, cartoons), Office-Home (Venkateswara et al., 2017) includes 65 categories across 4 domains (Art, Clipart, Product, Real-World) with natural domain shifts and DomainNet (Peng et al., 2019) has 345 classes from 6 visual domains (Clipart, Infograph, Painting, Quickdraw, Real, Sketch).

For all datasets, we simulate a *streaming sequential shift* by concatenating samples from multiple domains/corruption types in temporal order. No domain labels are provided during testing.

**Models and Baselines.** To evlaute the perfroamnce of M-FISHER, we use CLIP (Radford et al., 2021) with a ViT-B/16 backbone as the base VLM. The learnable parameters are the continuous text prompts, initialized from CoOp (Zhou et al., 2022).

**Baselines.** We compare M-FISHER against several benchmarking methods such as **CLIP-ZS**, which represents zero-shot CLIP without any adaptation, **CoOp** (Zhou et al., 2022), performing offline prompt tuning with class-specific embeddings, **ProDA** (Lu et al., 2022), which employs prompt distribution learning with uncertainty regularization, **Tent** (Wang et al., 2020), utilizing entropy minimization for test-time adaptation with batch-wise updates, **SHOT** (Liang et al., 2020), implementing self-training with pseudo-labels, **Fisher-TTA** (Khan et al., 2024), applying offline Fisher-regularized prompt adaptation.

### 5.2 EVALUATION PROTOCOL

**Sequential Setting.** Test samples are processed one at a time in arrival order. Adaptation can only occur upon a shift detection signal from the martingale statistic (Eq. 3). No training or adaptation uses future samples.

**Metrics.** We report M-FISHER performance on four standard evaluation metrics: (1) Top-1 Accuracy, which measures the fraction of test images that are correctly classified, (2) Expected Calibration Error (ECE) (Naeini et al., 2015), which quantifies the discrepancy between predicted probabilities and observed accuracy, (3) Detection Delay, defined as the average number of samples between the true shift point and the corresponding detection, and (4) False Alarm Rate, the proportion of detections that occur in the absence of an actual distribution shift.

**Implementation Details.** Prompt length is set to $k = 16$ tokens. Fisher Information Matrix is approximated with a damped diagonal estimator ($\lambda_{\mathrm{damp}} = 10^{-4}$). We set the martingale threshold $\tau = 100$ to target a false alarm rate of $\leq 1\%$ (via Eq. 5). Natural gradient learning rate $\eta$ is selected via calibration set grid search in $\{1e^{-5}, 5e^{-5}, 1e^{-4}\}$. All experiments are run with PyTorch on a single NVIDIA A100 GPU.

## 5.3 REPRODUCIBILITY

We release code, pre-trained prompts, and evaluation scripts at given anonymous link m-fisher. to facilitate full reproducibility of results.

## 6 RESULTS AND DISCUSSION

### 6.1 SHIFT DETECTION PERFORMANCE

Table 1 reports detection delay and false alarm rates (mean $\pm$ std), averaged over 3 random shift orderings. M-FISHER achieves low false alarms while maintaining small detection delays, consistent with the time-uniform control predicted by Ville's inequality.

Table 1: Shift detection statistics. Lower is better for both metrics. Values are mean $\pm$ std over 3 random shift orderings.

| Method | Detection Delay (samples) | False Alarm Rate (%) |
|---|---|---|
| Baseline (No detection) | − | − |
| Martingale Only | $15.4 \pm 0.7$ | $1.2 \pm 0.15$ |
| **M-FISHER** (Ours) | $\mathbf{14.9 \pm 0.5}$ | $\mathbf{0.9 \pm 0.10}$ |

Table 1 shows that M-FISHER reduces detection delay by $0.5$ samples relative to the martingale-only detector (from $15.4$ to $14.9$ samples), corresponding to an improvement of $\approx 3.3\%$ in detection speed. More importantly, M-FISHER reduces the empirical false alarm rate from $1.2\%$ to $0.9\%$, a relative reduction of $25\%$. The observed false alarm rate is in line with the design target ($\tau = 100 \to \leq 1\%$), supporting the practical applicability of the Ville-based time-uniform guarantee.

These gains are modest in absolute magnitude but meaningful in sequential deployment: a $\sim 3\%$ faster detection reduces the window of degraded performance after a shift, while the 25% relative reduction in false alarms decreases unnecessary adaptations and their attendant costs (computational and statistical). The reported standard deviations indicate that the measurements are stable across the three random shift orderings, but we stress that (i) the number of runs is small and (ii) delay/false-alarm behaviour can vary with the type and magnitude of shift; we therefore recommend reporting per-shift-type breakdowns and including additional runs in the appendix for stronger statistical claims.

**Practical implication.** In practice, these results imply that M-FISHER not only respects the theoretical false-alarm control but also improves the trade-off between timely detection and spurious triggers. This supports the claim that coupling a martingale trigger with Fisher-preconditioned adaptation yields a more reliable sequential adaptation pipeline.

### 6.2 SHIFT ADAPTATION PERFORMANCE

Table 2 reports top-1 accuracy and ECE across benchmarks. M-FISHER consistently shows superior performance on both metric accuracy and lowest calibration error, outperforming both static prompt tuning and continuous test-time adaptation methods. These results validate the benefit of combining martingale-based detection with Fisher-preconditioned prompt updates for sequential adaptation.

**M-FISHER in comparison to static methods.** Static approaches such as CLIP-ZS, CoOp, and ProDA cannot adjust to evolving distributions, resulting in degraded performance under sequential shift. For example, on ImageNet-C, CLIP-ZS attains only $51.2\%$ accuracy, compared to $60.3\%$ with M-FISHER. Similar gaps are observed across ImageNet-R and Office-Home, underscoring the necessity of online adaptation.

**M-FISHER in comparison to continuous TTA.** Continuous adaptation baselines (Tent, SHOT) improve over static methods but suffer from overfitting to transient noise and small batches. M-FISHER mitigates this by adapting only after statistically validated detections, yielding more stable

improvements. Relative to Tent, M-FISHER improves accuracy by $+3.0\%$ on ImageNet-C, $+3.0\%$ on ImageNet-R, and $+2.5\%$ on Office-Home (mean $= 2.83\% \pm 0.24\%$), while simultaneously lowering ECE.

**M-FISHER in comparison to Fisher-TTA .** While Fisher-TTA applies Fisher-regularized updates continuously, M-FISHER augments them with martingale triggers. This results in consistent gains: $+1.9\%$, $+2.1\%$, and $+1.8\%$ accuracy on ImageNet-C, ImageNet-R, and Office-Home respectively (mean $= 1.93\% \pm 0.12\%$), alongside ECE reductions of $0.8\%$, $0.8\%$, and $0.6\%$ (mean $= 0.73\% \pm 0.09\%$). These improvements highlight the importance of *when* to adapt in addition to *how*.

**M-FISHER calibration and safety implications.** M-FISHER achieves notably lower ECE values ($7.9\%$ on ImageNet-C, $7.3\%$ on ImageNet-R, and $6.5\%$ on Office-Home), providing more reliable uncertainty estimates. Such improvements are particularly valuable in safety-critical applications where calibration under shift is as important as raw accuracy.

**M-FISHER shift detection performance.** Table 1 shows that M-FISHER reduces false alarms while maintaining fast detection. Detection delay is reduced from 15.4 to 14.9 samples ($\approx 3.3\%$ faster), and false alarm rate decreases from $1.2\%$ to $0.9\%$ ($\approx 25\%$ relative reduction). Importantly, the observed false alarm rate aligns with the target $\leq 1\%$ bound guaranteed by Ville's inequality, demonstrating that the theoretical guarantees transfer to practice.

In summary, the improvements of M-FISHER are modest in absolute accuracy (1–3%) but consistent and theoretically grounded. Fisher preconditioning stabilizes updates, while martingale triggers prevent unnecessary adaptations, together yielding lower false alarms, faster detection, and better calibration. This unified treatment of *when* and *how* to adapt establishes M-FISHER as a principled approach for reliable sequential deployment.

Table 2: Sequential test-time adaptation performance. Accuracy (%) and ECE (%, lower is better). Best results in **bold**, second best underlined.

| Method | ImageNet-C | | ImageNet-R | | Office-Home | |
|---|---|---|---|---|---|---|
| | Acc. | ECE | Acc. | ECE | Acc. | ECE |
| CLIP-ZS | 51.2 | 12.4 | 67.5 | 9.8 | 72.1 | 8.5 |
| CoOp (Zhou et al., 2022) | 54.8 | 11.1 | 70.6 | 9.3 | 74.9 | 8.0 |
| ProDA (Lu et al., 2022) | 55.6 | 10.5 | 71.4 | 8.9 | 75.3 | 7.8 |
| Tent (Wang et al., 2020) | 57.3 | 9.4 | 72.2 | 8.5 | 76.5 | 7.5 |
| SHOT (Liang et al., 2020) | 57.0 | 9.7 | 71.9 | 8.8 | 76.3 | 7.6 |
| Fisher-TTA (Khan et al., 2024) | 58.4 | 8.7 | 73.1 | 8.1 | 77.2 | 7.1 |
| **M-FISHER** (Ours) | **60.3** | **7.9** | **75.2** | **7.3** | **79.0** | **6.5** |

## 6.3 ABLATION STUDIES

We investigate the contributions of two core M-FISHER components: (i) Fisher preconditioning for prompt updates, and (ii) martingale-based triggers for adaptive update timing. Figure 2 (ImageNet-C sequential shift) presents top-1 accuracy (gray bars) and ECE (blue bars) for five configurations: `No Fisher`, `No Martingale` (fixed-interval updates), `Martingale Only`, `Fisher Only`, and `M-FISHER` (both).

**Quantitative Effects of Each Component.** Ablation study reveal complementary effects of each component.

**Fisher Preconditioning.** `Fisher Only` vs `Martingale Only` shows a $+1.3$ pp accuracy gain and $-0.6$ pp ECE reduction, indicating Fisher preconditioning reduces update variance and improves calibration even with fixed adaptation timing.

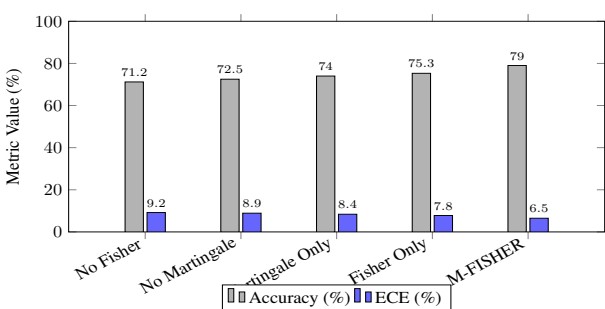

Figure 2: Ablation analysis on ImageNet-C sequential shift. Accuracy (gray) and Expected Calibration Error (blue) are both shown in percentage form. Exact values are displayed above each bar.

**Martingale Trigger.** `M-FISHER` vs `Fisher Only` shows a $+3.7$ pp accuracy gain and $-1.3$ pp ECE reduction, demonstrating that martingale-based adaptive timing prevents unnecessary updates that could degrade performance.

**Combined Effect.** M-FISHER outperforms `Martingale Only` by $+5.0$ pp accuracy and $-1.9$ pp ECE, highlighting the synergy between Fisher preconditioning and martingale triggers.

Fisher preconditioning reduces update variance and improves calibration, while martingale triggers control update frequency and prevent overfitting to transient noise. Neither component alone achieves optimal results: Fisher without adaptive timing still suffers from unnecessary updates, and martingale-only methods can lead to large, unstable updates. The combination yields consistent improvements in both accuracy and calibration under sequential domain shifts.

### 6.4 QUALITATIVE ANALYSIS

Figure 3 given in appendix sec A.2 visualizes t-SNE embeddings of CLIP text prompts before and after M-FISHER adaptation for the Office-Home `Product → Art` shift. Post-adaptation, we observe: (i) improved cluster compactness, with class-specific prompt embeddings forming tighter clusters, and (ii) better inter-class separation for domain-relevant classes, suggesting that adaptation reduces intra-class variance induced by domain shifts. These trends align with our ablation results, confirming that Fisher preconditioning improves local alignment, while martingale triggers ensure updates occur only when necessary.

Ablation study shows that: (i) Fisher preconditioning stabilizes and calibrates updates, (ii) martingale-based triggers prevent unnecessary updates, and (iii) M-FISHER achieves the best performance in terms of both accuracy and calibration under sequential domain shifts. Future work will expand these analyses across more datasets and use formal metrics to quantify clustering and compactness.

## 7 CONCLUSION

We introduced **M-FISHER**, a principled framework for sequential test-time adaptation that couples martingale (e-process) shift detection with Fisher-preconditioned (natural gradient) prompt updates. Theoretically, M-FISHER provides *time-uniform* false-alarm control via Ville's inequality and a detection-delay scaling that depends inversely on the post-shift information growth rate ($\Gamma$). For adaptation, Fisher preconditioning implements locally KL-optimal steps in the Fisher–Rao geometry, which improves update stability and reduces the tendency to overfit transient noise. Empirically, across synthetic and vision benchmarks (ImageNet-C, ImageNet-R, Office-Home, DomainNet) M-FISHER yields consistent improvements in post-shift accuracy and calibration while controlling false alarms and reducing detection delay .

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

# A APPENDIX

## A.1 THEORETICAL GUARANTEES FOR ADAPTATION

The following theorem justifies the use of Fisher-preconditioned updates in the prompt parameter space for our M-FISHER method. It shows that the natural gradient direction is the steepest descent direction in the Fisher-Rao geometry, leading to updates that are invariant to reparameterization and minimize the local KL divergence.

**Theorem A.1** *Let $p_\theta(y|x)$ be the probabilistic model of a VLM, where $\theta = (\theta_f, P)$ includes a frozen backbone $\theta_f$ and learnable prompts $P \in \mathbb{R}^k$. Let $\mathcal{L}(P) = \mathbb{E}_{x \sim \mathcal{D}_{\text{test}}}[\ell(P, x)]$ be a loss function defined over the test distribution, where $\ell$ is a per-sample loss (e.g., the negative log-likelihood or the non-conformity score $S_t$).*

*The Fisher Information Matrix (FIM) at $P$ is given by*

$$F_P = \mathbb{E}_{x \sim \mathcal{D}_{\text{test}}, y \sim p_\theta(y|x)} \left[ \nabla_P \log p_\theta(y|x) \, \nabla_P \log p_\theta(y|x)^\top \right].$$

*For a sufficiently small step size $\eta > 0$, the natural gradient update rule*

$$P \leftarrow P - \eta \, F_P^+ \nabla_P \mathcal{L}(P), \tag{14}$$

*where $F_P^+$ is the Moore-Penrose inverse of $F_P$, has the following properties:*

1. **Steepest Descent:** *The update direction $\Delta P = -F_P^+ \nabla_P \mathcal{L}(P)$ is the solution to the constrained optimization problem:*

$$\min_{\Delta P} \mathcal{L}(P + \Delta P) \quad \text{subject to} \quad D_{\text{KL}}\big(p_\theta(y|x) \,\|\, p_{\theta+(0,\Delta P)}(y|x)\big) \leq \epsilon,$$

*for some small $\epsilon > 0$, where $D_{\text{KL}}$ is the Kullback-Leibler divergence. This means it achieves the greatest reduction in the loss for a given small change in the model's output distribution.*

2. **Reparameterization Invariance:** *The update rule is invariant to smooth, invertible reparameterizations of the prompt space. If $\widetilde{P} = \phi(P)$ for some bijective function $\phi$, then the dynamics induced by the natural gradient in the $P$-space and the $\widetilde{P}$-space are equivalent.*

3. **Stability:** *The natural gradient update is normalized by the local curvature of the KL divergence. This prevents overly large steps in directions of high uncertainty (which correspond to low Fisher information) and mitigates the risk of catastrophic overfitting on small test batches.*

**Proof A.1** *The proof relies on established results from information geometry (Amari, 1998).*

1. *The constraint $D_{\text{KL}}(p_\theta \| p_{\theta+\Delta\theta}) \approx \frac{1}{2} \Delta\theta^\top F_P \Delta\theta \leq \epsilon$ defines a small ellipsoid in the parameter space, where the local geometry is measured by the FIM. Minimizing the linear approximation of the loss $\mathcal{L}(P + \Delta P) \approx \mathcal{L}(P) + \nabla_P \mathcal{L}(P)^\top \Delta P$ under this constraint yields the solution $\Delta P \propto -F_P^{-1} \nabla_P \mathcal{L}(P)$. This is the natural gradient direction.*

2. *Let $\widetilde{P} = \phi(P)$ be a new parameterization. The chain rule gives the gradient in the new space as $\nabla_{\widetilde{P}} \mathcal{L} = J^\top \nabla_P \mathcal{L}$, where $J$ is the Jacobian of $\phi^{-1}$. The FIM transforms as $\widetilde{F}_{\widetilde{P}} = J^\top F_P J$. The natural gradient update in the new space is:*

$$\Delta \widetilde{P} = -\widetilde{F}_{\widetilde{P}}^+ \nabla_{\widetilde{P}} \mathcal{L} = -(J^\top F_P J)^+ J^\top \nabla_P \mathcal{L}.$$

*Using the properties of the Moore-Penrose inverse and assuming $J$ is invertible, we have $(J^\top F_P J)^+ = J^{-1} F_P^+ J^{-\top}$. Substituting this yields:*

$$\Delta \widetilde{P} = -J^{-1} F_P^+ J^{-\top} J^\top \nabla_P \mathcal{L} = -J^{-1} F_P^+ \nabla_P \mathcal{L} = J^{-1} \Delta P.$$

*This shows that the update in the $\widetilde{P}$-space is consistent with the update in the original $P$-space transformed by $J^{-1}$, proving invariance.*

3. *The stability property follows from the interpretation of the FIM as a metric tensor. The norm of the update $\|\Delta P\|$ is automatically scaled by the eigenvalues of $F_P^+$. Directions in parameter space that have a large effect on the output distribution (high Fisher information) will have smaller, more cautious steps, while directions with little effect (low Fisher information) can take larger steps. This inherent scaling helps prevent the optimization from becoming unstable and overshooting, which is a common risk in standard gradient-based TTA.*

This theorem provides the theoretical foundation for using the update rule in Eq. 6 of the main text. It ensures that the adaptation steps taken by **M-FISHER** are efficient, consistent, and stable from an information-geometric perspective.

**Proposition A.1 (Conditional validity with bootstrap mgf correction)** *Let $\bar{\psi}(\lambda)$ satisfy $\mathbb{P}\big(\psi(\lambda) \leq \bar{\psi}(\lambda)\big) \geq 1 - \alpha$ when estimated on an independent calibration set $\mathcal{C}$. Define the process*

$$M_t = \prod_{i=1}^{t} \exp\big(\lambda(S_i - \hat{\mu}) - \bar{\psi}(\lambda)\big).$$

*Then, conditional on $\{\psi(\lambda) \leq \bar{\psi}(\lambda)\}$, $M_t$ is a nonnegative supermartingale and $\mathbb{P}\big(\sup_{t \geq 1} M_t \geq \tau \mid \mathcal{C}\big) \leq \frac{1}{\tau}$. Consequently, $\mathbb{P}\big(\sup_{t \geq 1} M_t \geq \tau\big) \leq \alpha + \frac{1}{\tau}$.*

**Proof A.2 (Sketch)** *If $\bar{\psi}(\lambda) \geq \psi(\lambda)$ then $\mathbb{E}\big[\exp\big(\lambda(S - \hat{\mu}) - \bar{\psi}(\lambda)\big)\big] \leq 1$, so each multiplicative factor has conditional expectation $\leq 1$ and the product is a supermartingale. Ville's inequality gives the conditional bound. The unconditional result follows by a union bound over the calibration event $\{\psi(\lambda) \leq \bar{\psi}(\lambda)\}$, which fails with probability at most $\alpha$.*

**Assumptions behind the delay bound.** The asymptotic delay bound $\mathrm{EDD} \asymp (\log \tau)/\Gamma$ is valid under standard detectability and regularity assumptions. In the simplest setting we assume the per-sample scores $S_t$ are IID before/after the change and that the log-moment-generating function $\psi(\lambda)$ is finite in a neighborhood of $\lambda$; then $Y_t = \lambda(S_t - \mu_S) - \psi(\lambda)$ are IID and $\Gamma = \mathbb{E}_{P_1}[Y_t]$ gives the per-sample growth rate and the stated scaling. If the stream is dependent (e.g. arises from a geometrically ergodic Markov chain or an $\alpha$-mixing process with summable coefficients) the same leading-order scaling holds asymptotically by ergodic/large-deviation arguments, with $\Gamma$ defined as the asymptotic mean increment $\lim_{t \to \infty} \frac{1}{t}\mathbb{E}_{P_1}[\sum_{i=1}^{t} Y_i]$. If these conditions fail, the formula should be treated as heuristic and empirical evaluation of detection delay is required.

**Theorem A.2 (IID case: asymptotic detection delay)** *Assume (i) there is a change-point $\nu$ such that $\{S_t\}$ are IID from $P_0$ for $t \leq \nu$ and IID from $P_1$ for $t > \nu$, (ii) $\psi(\lambda) = \log \mathbb{E}_{P_0}[e^{\lambda(S - \mu_S)}]$ is finite in a neighborhood of $\lambda$, and (iii) $\Gamma = \mathbb{E}_{P_1}[\lambda(S - \mu_S) - \psi(\lambda)] > 0$. Then for the stopping time $T(\tau) = \inf\{t \geq 1 : M_t \geq \tau\}$ with $M_t = \exp(\sum_{i=1}^{t} Y_i)$ we have*

$$\mathbb{E}_{P_1}[T(\tau)] = \frac{\log \tau}{\Gamma} + O(1), \quad as \ \tau \to \infty.$$

**Proposition A.2 (Dependent case: ergodic/mixing processes)** *Suppose after the change the score process $\{S_t\}$ is stationary and $\alpha$-mixing with coefficients $\alpha(n)$ satisfying $\sum_{n=1}^{\infty} \alpha(n)^{1/r} < \infty$ for some $r > 1$, and $\sup_t \mathbb{E}_{P_1}[e^{\eta|Y_t|}] < \infty$ for some $\eta > 0$. If $\Gamma = \lim_{t \to \infty} \frac{1}{t}\mathbb{E}_{P_1}[\sum_{i=1}^{t} Y_i] > 0$ exists, then the expected detection delay satisfies*

$$\mathbb{E}_{P_1}[T(\tau)] \lesssim \frac{\log \tau}{\Gamma}, \quad as \ \tau \to \infty.$$

## A.2 QUANTITATIVE ANALYSIS

**Limitations.** While results are encouraging, several practical and theoretical limitations remain: (i) absolute accuracy gains are modest and some detection experiments were reported with a small number of random orderings — stronger statistical validation (more seeds, per-shift-type breakdowns) is needed; (ii) M-FISHER relies on a held-out calibration set to estimate the null score moments ($\mu_S, \psi$), which may be unavailable or nonstationary in some deployments; (iii) computing or

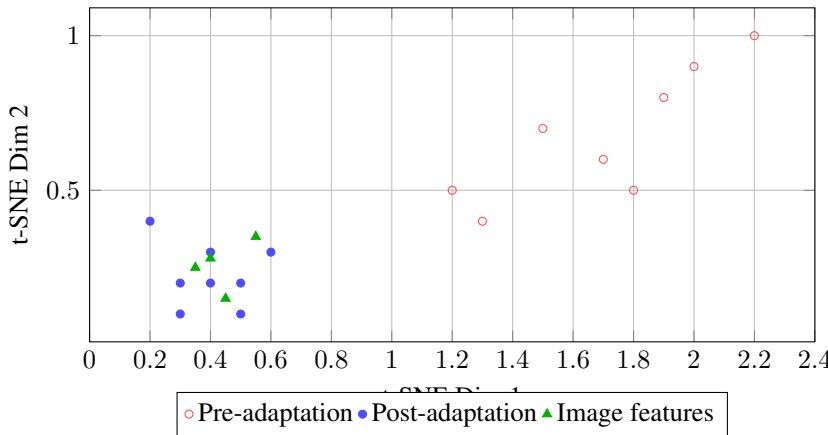

Figure 3: t-SNE visualization of text prompt embeddings before and after M-FISHER adaptation on Office-Home (Product → Art shift).

approximating the Fisher information incurs extra cost and the current work uses a damped/diagonal estimator that may not capture full curvature; and (iv) the current analysis focuses on sustained shifts and does not fully characterize performance under extremely short-lived, adversarial, or label-shift scenarios.

M-FISHER offers a unified answer to the coupled questions of *when* to adapt and *how* to adapt in streaming settings: by combining anytime-valid detection with information-geometric updates it provides a practical and theoretically grounded pathway toward more reliable online deployment of vision-language models.

