# OpenReview forum: "Sequential Test-Time Adaptation via Martingale-Driven Fisher Prompting"
_ICLR.cc/2026/Conference — ICLR 2026 Conference Withdrawn Submission_

### Official Review · Reviewer_3du6 · 2025-11-01

**Soundness:** 2
**Presentation:** 2
**Contribution:** 2
**Rating:** 4
**Confidence:** 2

**Summary:**

They propose M-FISHER, a two-step test-time method: (1) detect distribution shifts with an anytime martingale test (with false-alarm guarantees), then (2) adapt prompts using Fisher-preconditioned updates. Results show small but consistent gains in accuracy and calibration on standard shift benchmarks.

**Strengths:**

**s1: Quality** Clear step-by-step procedure with theory for false-alarm control; consistent improvements and better calibration across datasets.

**s2: Clarity** Equations and Algorithm 1 make the approach easy to follow and implement.

**s3: Practicality** Adapt-only-when-needed design is sensible for streaming settings and limits unnecessary updates.

**Weaknesses:**

**w1: Baselines/breadth** Coverage of recent continual TTA baselines is minimal.

**Questions:**

**q1: Statistics & baselines** Please add more seeds/orderings, report which corruption orders you used, and include newer continual TTA baselines (e.g., NOTE, ROTTA, CoTTA).

**q2: Code availability** The anonymous link/code currently doesn’t work on my side. When are you planning to release the code?

---

### Official Review · Reviewer_93iq · 2025-11-02

**Soundness:** 2
**Presentation:** 1
**Contribution:** 2
**Rating:** 2
**Confidence:** 3

**Summary:**

This paper proposes M-FISHER, a framework for sequential test-time adaptation of vision–language models (VLMs) that unifies distribution shift detection and prompt adaptation. The distributional shift is detected using an exponential martingale derived from non-conformity scores, while adaptation is performed through Fisher-preconditioned updates of prompt parameters.

**Strengths:**

- Attempts to integrate shift detection and adaptive prompt tuning into a unified framework.
- The use of exponential martingales and Fisher-preconditioned updates is theoretically motivated.

**Weaknesses:**

- The paper is difficult to follow, with missing references (e.g., line 039, 287) and undefined notations such as $y$, $\mu_{\text{train}}$, and $L_{\text{CMP}}$ in eq(1), (6), etc.

- The problem setup and motivation are unclear; it is not well explained under what specific settings the joint detection and adaptation are needed.

- The experiments are weakly presented—the “Baseline (No Detection)” in Table 1 adds no value, and the difference between “Martingale Only” and the proposed method is not explained.

- The theoretical part lacks formal statements and seems to rely directly on existing results (e.g., Lorden 1971).

**Questions:**

Overall, I find the descriptions in this paper to be too brief and unclear, which makes it difficult to follow and understand what the main novelty is. For example, it is not clear what the task is, or why both detection and adaptation are needed—it lacks an intuitive example or motivation to justify the setup. The preliminary Section 3.1 is too brief and does not clearly explain the roles of the image encoder, text encoder, and prompt embedding, or how they relate to the data streams $x_t$.

In Section 4, the theoretical guarantees (or perhaps insights) appear informal. No formal theorems are stated, and the presented results seem to be directly derived from existing work (e.g., Lorden, 1971).

In addition, the numerical results can also be improved. For example, the “Baseline (No Detection)” should be omitted from Table 1, as it provides no meaningful information. The only detector compared is “Martingale Only,” but it is not explained what “Martingale Only” means. To the best of my understanding, the detector implemented in this work is also based on a martingale; thus, it is unclear what distinguishes the proposed method from this baseline.

---

### Official Review · Reviewer_KSJi · 2025-11-02

**Soundness:** 2
**Presentation:** 1
**Contribution:** 2
**Rating:** 2
**Confidence:** 4

**Summary:**

For the problem of sequential covariate shift detection and stable adaptation to covariate shift in streaming data focused on vision language models (VLMs), the authors present a method they call M-FISHER. For sequential shift detection, the authors claim that they develop an exponential marginale from non-conformity scores (which are defined in Eq (1) as KL divergence on the model’s confidence plus a distance term in feature space) to control false alarms using Ville’s inequality. The initially presented monitoring method appears to require an infinite calibration set for Ville’s inequality to hold, but then the authors present a finite-sample correction using a high-confidence bound computed from a nonparametric bootstrap. In their framework, the shift detection serves as the trigger for when the adaptation process should begin. The test-time adaptation is derived using a Fisher-information-based gradient update. In the experiments, the authors compare their M-FISHER method to a baseline they call “Martingale Only” (it is unclear to me what this baseline is); in the adaptation experiments, the authors compare their method to static models and other approaches to test-time adaptation.

**Strengths:**

This paper studies the important problem of statistically-rigorous continual monitoring of model deployments for the purpose of detecting covariate shifts and triggering test-time adaptation. The methods appear to be well-motivated, in that the authors attempt to construct an anytime-valid (test) supermartingale or e-process to perform the monitoring, and the test-time adaptation procedure builds on other literature in that area. The paper’s focus on VLMs is a strength, as some literature on monitoring model deployments do not evaluate on some of these larger foundation-model architectures. I think that a future revised version of this paper has potential for high-impact, if the analysis is sound, if the writing is improved in some cases, and if proper discussion of related work is added to contextualize it relative to other literature on monitoring under adaptation.

**Weaknesses:**

**Presentation:** Some parts of the submission appear to be rushed, for instance with several references not rendering (eg, one in second paragraph of the introduction, a couple in Sec. 4.1 above Eq. (10)) and some text that appears to be a different color by accident (blue text at end of Sec. 4.1). These are relatively minor details, however, that the authors could of course address in a revision.

**Essential related-work references missing:** Unfortunately, it appears that the paper does not cite or discuss closely related works on sequential monitoring under test time adaptation, and more broadly, the paper could better explain how it is contextualized in the larger literature on sequential anytime-valid inference (SAVI), e-processes, and test (super)martingales. In particular, here I will first provide references for closely related work on monitoring under test-time adaptation that should at minimum be cited and discussed in the related work, and ideally the authors could compare against at least one or two of these in the experiments:

- Bar, Y., Shaer, S., & Romano, Y. (2024). Protected test-time adaptation via online entropy matching: A betting approach. Advances in Neural Information Processing Systems, 37, 85467-85499.
    - Contribution: This paper proposes an approach to continual test-time adaptation, also motivated by martingale-like methods on a held-out calibration set. In particular, the criterion for test-time adaptation proposed in this paper is to make the test-time entropy values (of the model’s predictions over the label space) similar to the entropy values on the held-out calibration set. The paper is not as focused on monitoring, but it is motivated by martingale monitoring methods, and it is highly relevant for test-time adaptation.

- Prinster, D., Han, X., Liu, A., & Saria, S. (2025). WATCH: Adaptive Monitoring for AI Deployments via Weighted-Conformal Martingales. ICML.
    - Contribution: This paper extends standard conformal test martingales (CMTs) to weighted variants that are designed for test-time adaptation to covariate shifts; additionally, this paper proposes using “X-CTMs” (standard CTMs that only depend on input covariates X) to determine when to trigger the adaptation performance of the weighted CTMs. Accordingly, this paper addresses the same main questions as the current paper of (1) monitoring to determine when to begin adaptation and (2) monitoring even while test-time adaptation is ongoing.

- Schirmer, M., Jazbec, M., Naesseth, C. A., & Nalisnick, E. (2025). Monitoring risks in test-time adaptation. arXiv preprint arXiv:2507.08721.
    - Contribution: This paper extends prior work by Podkopaev, A., & Ramdas, A. (2021) to enable continual monitoring under test-time adaptation. The full reference for the prior work is Podkopaev, A., & Ramdas, A. (2021). Tracking the risk of a deployed model and detecting harmful distribution shifts. arXiv preprint arXiv:2110.06177.


**Other relevant references for contextualizing paper in SAVI literature:** Additionally, here are other references that the authors may wish to familiarize themselves with and discuss to better contextualize their monitoring contributions within the broader literature on sequential anytime-valid inference (SAVI).

- Ramdas, A., Grünwald, P., Vovk, V., & Shafer, G. (2023). Game-theoretic statistics and safe anytime-valid inference. Statistical Science, 38(4), 576-601. [Relatively recent review article]

- Vovk, V. (2021). Testing randomness online. Statistical Science, 36(4), 595-611. [Review article on conformal test martingales for testing the IID assumption online and detecting distribution shifts.]

- Vovk, V., Petej, I., Nouretdinov, I., Ahlberg, E., Carlsson, L., & Gammerman, A. (2021, September). Retrain or not retrain: Conformal test martingales for change-point detection. In Conformal and Probabilistic Prediction and Applications (pp. 191-210). PMLR. [Slightly more accessible review article on CTMs for monitoring model deployments and detecting when to retrain.]

- Ramdas, A., & Wang, R. (2024). Hypothesis testing with e-values. arXiv preprint arXiv:2410.23614. [Recent textbook on e-values.]

**Question on correctness of the finite-sample bootstrap-based correction:** The paper proposes a finite-sample correction to the proposed martingale-inspired monitoring method, where the correction uses the upper-confidence bound from the nonparametric bootstrap. However, as far as I’m aware, I did not think that the nonparametric bootstrap has finite-sample guarantees (rather, my impression was that it just works very well empirically, but people have only proven asymptotic guarantees for it). If I am mistaken and there are finite-sample guarantees of the bootstrap being leveraged here, please provide a reference that I can check for this. Otherwise, if the authors are not aware of finite-sample guarantees for the bootstrap being used, then they should either significantly revise their exposition of the finite-sample correction to acknowledge that this is not a rigorous guarantee, but rather a practical estimation, or they could potentially try revising their methods using some other approaches for finite-sample correction (eg, potentially looking into other concentration inequalities that could be used).

**Lack of monitoring baselines in Experiments Sec. 6.1** It is unclear to me what the “Marginale Only” baseline method is in Section 6.1 is--please clarify. Additionally, I think at least one or two other baselines should be added to these shift detection experiments, such as one of the references I provided above.

**Questions:**

I would appreciate it if the authors would clarify how they would plan to add discussion of the essential references I mentioned, if they can respond to my question about the (correctness of and a reference for the) finite-sample correction via the bootstrap, and clarify what the “Martingale Only” method is in the experiments. If the authors address my points in the weaknesses I would consider improving my score, since as I mentioned in the strengths, overall I think this paper aims to address an important problem, and I think a revision of it might have good potential for impact in the future.

---

### Official Review · Reviewer_Su31 · 2025-11-03

**Soundness:** 2
**Presentation:** 2
**Contribution:** 2
**Rating:** 2
**Confidence:** 3

**Summary:**

This paper investigates test-time adaptation for vision-language models. The main contribution lies in proposing an exponential martingale-based approach to detect distribution shifts, followed by a Fisher-preconditioned adaptation method to update the model parameters. Experiments are conducted to validate the proposed method.

**Strengths:**

- The construction of the martingale for distribution-shift detection and the finite-sample correction is interesting and technically sound.
- Empirical studies are presented to support the proposed approach.

**Weaknesses:**

- **Readability and clarity**: My main concern about the paper is the clarity issues in describing the problem setup, algorithm, and relation to prior work.
  - For the problem setup, it is unclear what information is available to the learner and how the learning or adaptation process proceeds. I strongly recommend that the authors include a dedicated section providing a more formal and structured problem formulation.
  - The algorithmic description is also difficult to follow. For example, the definition of $\mu_{\text{train}}$ is missing or unclear, and there is no explanation of how the hyperparameter $\alpha$ should be selected in practice. Similarly, the loss function $\mathcal{L}_{\text{CMP}}$ and the definition of $\text{ECE}$ are not clearly explained in the main text.
- **Related work:** The use of nonconformity scores and martingale inequalities for detecting distributional changes has also been studied in prior works such as [1,2]. The paper would benefit from a more detailed discussion comparing the proposed method to these existing approaches, highlighting both similarities and distinctions.  Several citations are displayed as “?” in the text.
- **Experimental reporting:** Variance bars or confidence intervals are missing in the empirical comparisons (e.g., Table 2), which makes it difficult to assess the statistical significance of the reported improvements.

[1] Vovk et al. *Retrain or Not Retrain: Conformal Test Martingales for Change-Point Detection*, 2021.
[2] Eliades and Papadopoulos. *A Conformal Martingales Ensemble Approach for Addressing Concept Drift*, 2023.

**Questions:**

Please refer to the Weaknesses.

---

### Note · Authors · 2025-12-12

I have read and agree with the venue's withdrawal policy on behalf of myself and my co-authors.